# Exploring the Efficacy of Vessilen^®^ in Treating Bladder Pain Syndrome/Interstitial Cystitis: A Prospective Study

**DOI:** 10.3390/healthcare13111340

**Published:** 2025-06-04

**Authors:** Mariachiara Palucci, Marta Barba, Alice Cola, Matteo Frigerio

**Affiliations:** Department of Gynecology, Fondazione IRCCS San Gerardo dei Tintori, University of Milano-Bicocca, 20900 Monza, Italy; m.barba8792@gmail.com (M.B.); alice.cola1@gmail.com (A.C.); frigerio86@gmail.com (M.F.)

**Keywords:** bladder pain syndrome, interstitial cystitis, IC/BPS, Adelmidrol, intravesical treatment, Vessilen^®^

## Abstract

**Background/Objectives:** Bladder pain syndrome (BPS), or painful bladder syndrome (PBS)/interstitial cystitis (IC), is a chronic inflammatory condition characterized by symptoms like pain, urgency, urinary incontinence, and sometimes urinary retention, which significantly affect patients’ quality of life. The etiology of PBS/IC remains unclear and may be multifactorial, with no definitive treatment currently available. The challenge lies in finding new therapeutic strategies. Various intravesical treatments, such as heparin, hyaluronic acid, and botulinum toxin, are commonly used for PBS/IC. In this study, we aimed to evaluate the anti-inflammatory effects of intravesical Vessilen^®^ (a new formulation consisting of 2% adelmidrol and 0.1% sodium hyaluronate) in patients with IC/PBS or other bladder disorders. **Methods:** This was a pilot study conducted at a tertiary-level urogynecology center. Two validated questionnaires were administered to patients before and after treatment: the Visual Analogue Scale (VAS) and the International Consultation on Incontinence Questionnaire Female Lower Urinary Tract Symptoms Modules (ICIQ-FLUTS Long Form). The Patient Global Impression (PGI) scale was used to assess symptom severity. **Results:** Among the 25 patients who completed six weekly instillations, a significant decrease in bladder symptoms was observed, as indicated by both the ICIQ-FLUTS scale (89.3 vs. 61.3; *p* = 0.021) and VAS score (4.4 vs. 2.6; *p* < 0.001). Additionally, 80% of patients reported symptom improvement (PGI-I score ≤ 3). **Conclusions:** Intravesical Vessilen^®^ (adelmidrol + sodium hyaluronate) appears to be an innovative therapeutic approach for PBS/IC and other chronic inflammatory bladder disorders due to its anti-inflammatory and antinociceptive properties.

## 1. Introduction

Bladder pain syndrome (BPS)/interstitial cystitis (IC) is a clinical inflammatory condition characterized by chronic pain originating from the bladder and/or pelvis, accompanied by urinary urgency or frequency, in the absence of other identifiable causes. The International Continence Society (ICS) prefers the term “painful bladder syndrome (PBS)”, defined as “suprapubic pain related to bladder filling, accompanied by other symptoms such as increased daytime and nighttime frequency, in the absence of proven urinary infection or other obvious pathology” [1]. The historical definition provided by the National Institute of Diabetes and Digestive and Kidney Diseases (NIDDK) in 1987 and revised in 1988 required the presence of either glomerulations or “Hunner ulcers” upon cystoscopic examination [2], whereas contemporary urological societies, including the American Urological Association (AUA), the European Society for the Study of Interstitial Cystitis (ESSIC), and ICS, have moved away from the requirement for cystoscopy with hydrodistension as a necessary diagnostic criterion [3,4]. Although the diagnostic definition has been modified over time, becoming less restrictive regarding the duration of symptoms and the cystoscopic findings, a unified consensus has yet to be established.

As previously mentioned, excluding confusable diseases presenting similar symptoms is mandatory; however, differentiating IC/PBS from chronic urinary tract infections (UTIs) [5] is particularly challenging due to potential overlapping mechanisms and the possibility of coexistence between the two conditions [6].

Prevalence estimates for IC/PBS have increased as diagnostic criteria have evolved. Earlier reports suggested a range of 10 to 510 cases per 100,000 individuals. A population-based cross-sectional study conducted by Clemens et al. in Boston found that the prevalence of IC/PBS varied according to the definition used, being 0.83% among women when a more restrictive definition was used versus 2.71% when a more inclusive one was employed [7]. However, the condition may be underreported, with fewer than 10% of individuals with IC/PBS receiving a formal diagnosis [8].

The exact pathogenesis of PBS remains unclear, though a multifactorial origin involving interactions between the bladder epithelium, immune system, neural pathways, and the microbiome is widely considered the most plausible hypothesis [9]. Consequently, PBS is often difficult to manage, and a stepwise, multimodal approach is mandatory [10].

PBS/IC may be conceptualized as occurring in two phases: the first phase involves processes that lead to epithelial cell dysfunction, including increased permeability and the generation of inflammatory or immune responses. The second phase is characterized by progressive tissue and nerve changes in the bladder, resulting in chronic pain and the characteristic storage symptoms [9]. In the course of PBS, the protective function of the urothelial coating is progressively compromised by the phenomenon of transepithelial mast cell superficialization, which induces local neuroinflammation and the release of depolymerizing enzymes (proteases and hyaluronidase) that target the fundamental components of the urothelial coating (glycosaminoglycans and GAGs) [11,12].

Furthermore, dysregulated mast cell activation has been found, especially in areas with weakened urothelium or in the presence of typical Hunner’s lesions [13,14].

Infection could trigger the morphological alterations seen in IC/PBS since evidence links decreased antibacterial glycoprotein levels and hyperactivation of immune cells with this condition.

As a painful condition, IC/PBS exhibits neural pathophysiological mechanisms commonly seen in other chronic algic syndromes, such as enhanced neuronal excitability and central nervous system remodeling [15,16], which play a key role in symptom persistence.

As anticipated, there is ongoing debate regarding the importance of cystoscopic findings, such as Hunner’s lesions (seen in less than 10% of IC/PBS patients) and glomerulations (mucosal bleeding after bladder overdistension), in relation to symptoms [17]. The significant improvement observed after the resection of Hunner’s lesions, along with their documented association with pain and urgency, and the finding that patients without Hunner’s lesions have a statistically higher prevalence of chronic disorders have led some to suggest there are two distinct categories of interstitial cystitis/bladder pain syndrome (IC/BPS): ulcerative IC/PBS and non-ulcerative IC/PBS [18]. Clinically, the former is characterized by an older age of onset, more severe bladder-centric symptoms, diminished bladder capacity, fewer comorbid non-bladder syndromes, and more-favorable outcomes upon endoscopic treatment.

Although glomerulations are a very common finding in this condition, affecting approximately 90% of patients according to Simon et al.’s preliminary data from the Interstitial Cystitis Data Base (ICDB), a strong correlation between cystoscopic findings (the presence, density, and diffuseness of glomerulations) and the severity of urinary symptoms was not found [19]; additionally, Wennevik and colleagues suggested that glomerulations are not pathognomonic for IC/PBS [20]. However, due to the discordant results in the literature, no definitive conclusion can be made regarding the relationship between cystoscopic findings and symptoms.

Considering the complexity of the condition, a multimodal treatment approach is essential.

Although the best treatment has not yet been identified, an appropriate approach should always start with conservative therapies, progressing to more aggressive options if symptom control is insufficient or if the patient’s quality of life (QoL) remains poor. Furthermore, the key to successful outcomes could lie in the combination of pharmacological treatments and manual physical therapy techniques, which are particularly recommended for patients experiencing pelvic floor tenderness [21].

Oral medications such as amitriptyline, cimetidine, hydroxyzine, and pentosan polysulfate (Elmiron^®^) have been used to alleviate IC/PBS symptoms, primarily serving as supplements to other treatments [21].

Moving on to more invasive treatments, delivering medication directly to the bladder is a logical strategy to control local inflammation. Intravesical instillation has become a standard treatment for IC/BPS due to several benefits: (i) higher drug concentrations in the bladder, (ii) fewer systemic side effects, (iii) a reduced risk of drug interactions compared to oral medications, and (iv) direct repair of urothelial defects [22].

However, intravesical drug delivery has some limitations. The impermeability of urothelial cells, characterized by tight junctions and umbrella cells, can hinder treatment efficacy. Additionally, the short duration of action and the need for frequent administration can induce pain, be costly, and potentially increase the risk of infection.

Given the multifactorial nature of this disease, the intravesical instillation of drugs containing different molecules may enhance therapeutic outcomes. Several compounds have been tested so far, including heparin, heparinoids (such as hyaluronic acid and chondroitin sulfate), pentosan polysulfate (Elmiron^®^), dimethylsulfoxide (DMSO), and botulinum toxin A (BoNT-A) [23].

In this context, Aldemidrol, an analogue of palmitoylethanolamide (PEA), may represent a valid alternative for an intravesical approach. It is a well-known anti-inflammatory and antioxidant compound used for the management of acute and chronic inflammation. Specifically, Aldemidrol is an innovative active product able to control the mast cell component through the ALIA (Autacoid Local Injury Antagonism) mechanism. It makes this possible both by directly modulating mast cell activity through the activation of the nuclear receptor PPAR-γ and by increasing local endogenous levels of palmitoylethanolamide [23].

The present study was designed as the first attempt to evaluate the anti-inflammatory effects of intravesical Vessilen^®^ (a new formulation consisting of 2% adelmidrol (the diethanolamide derivative of azelaic acid) + 0.1% sodium hyaluronate) administration on patients affected by IC/PBS or with conditions associated with local inflammation, voiding dysfunctions, and pain in the pelvic/perineal area.

## 2. Materials and Methods

A pilot study was conducted at a tertiary-level urogynecology center (San Gerardo Hospital, Monza, Italy) between November 2023 and July 2024. All patients deemed eligible for the study reported chronic pelvic pain associated with lower-urinary-tract symptoms, including increased urinary frequency, urgency, and unpleasant pressure and discomfort. Recruitment was limited to patients with symptoms lasting for at least 3 to 6 months.

The enrolled patients received a treatment regimen consisting of six intravesical instillations of Vessilen^®^ (a novel formulation containing 2% adelmidrol, the diethanolamide derivative of azelaic acid, and 0.1% sodium hyaluronate), administered weekly over a period of six weeks.

Before being referred for treatment, the patients underwent a urogynecological examination, which included a detailed assessment of urinary symptoms, a pelvic ultrasound, and a urine culture with a negative result. Hospitalization was not necessary for carrying out the procedure.

After placing each patient in the gynecological position and disinfecting the genital area, we gently inserted a small-caliber urinary catheter and administered 50 mL of pure Vessilen^®^ preparation using a syringe connected to the catheter. No dilution or reconstitution of the drug was necessary before instillation.

The following validated questionnaires were administered to the patients both before and after the treatment to assess the outcomes: the Visual Analogue Scale (VAS) and the International Consultation on Incontinence Questionnaire Female Lower Urinary Tract Symptoms Module (ICIQ-FLUTS Long Form) [24,25]. The VAS is a scale used to measure the intensity of symptoms as perceived by the patient, ranging from 0 (no symptoms) to 10 (the worst possible intensity). The ICIQ-FLUTS is used to evaluate 18 items related to female lower-urinary-tract symptoms and their impact on quality of life (QoL) in both research and clinical practice worldwide. In addition, personal satisfaction was assessed using the Patient Global Impression Scale (PGI-I), which was completed by the patients after the last instillation.

The study protocol was reviewed and approved by our institution’s ethics committee (approval N° 185/CE, 24 February 2022; protocol code RE-PFDS). All patients voluntarily agreed to participate in the study and provided their written informed consent.

All patients were carefully monitored during the treatment period for any adverse effects. Anonymized data were prospectively collected by the authors. Statistical analysis was performed using *JMP software version 9 (SAS Institute, Cary, NC, USA)*. Outcomes were reported as means ± standard deviations for continuous variables and absolute or relative frequencies for non-continuous variables. Pre- and post-treatment comparisons were made for both objective and subjective outcomes and tested for statistical significance. Differences were assessed using a paired *t*-test for continuous data and Fisher’s exact test for non-continuous data. A *p*-value of <0.05 was considered statistically significant.

## 3. Results

A total of 26 patients were enrolled. The average age was 58.4, with the youngest patient being 32 years old and the oldest being 87 years old. Over 80% of the patients had undergone at least one vaginal delivery, while 11 (42.3%) had undergone previous pelvic surgeries, five of which were for urinary incontinence or pelvic organ prolapse. Smoking was only an exacerbating risk factor for one patient.

The most common symptom presented was increased urinary frequency (14 out of 25 patients, 56%), followed by urinary urgency with occasional episodes of incontinence (8 out of 25 patients, 32%).

The population’s baseline characteristics are summarized in Table 1.

A total of 25 patients (96.15%) successfully completed all six planned intravesical instillations of Vessilen^®^. One patient (3.85%) chose to withdraw from the protocol after the first instillation due to poor tolerance of urethral catheterization and was therefore excluded from the analysis. The mean scores of the pre- and post-treatment questionnaires are reported in Table 2. We observed a significant decrease in the severity of bladder symptoms according to both the total ICIQ-FLUTS scale (89.3 vs. 61.3; *p* = 0.021) and the VAS score (4.4 vs. 2.6; *p* < 0.001). More specifically, a significant score reduction was observed for 15 of the 18 items comprising the ICIQ-FLUTS questionnaire, as evidenced in Table 2 and Figure 1. Moreover, according to PGI-I, 80% of the patients noted an improvement in symptoms (PGI-I score ≤ 3).

## 4. Discussion

IC/PBS is a chronic condition characterized by recurrent pelvic pain or pressure and increased urinary frequency. Similarly, women with recurrent cystitis may also experience the same symptoms, despite prolonged pharmacological treatments and supplements. These medical issues significantly deteriorate both mental and physical quality of life, often leading to depression, anxiety, and reduced social interactions [26].

Despite having undergone revisions over time, the diagnostic approach has shifted toward identifying the presence of the aforementioned symptom constellation, with cystoscopy being recommended only when deemed necessary to exclude differential diagnoses [27].

The exact cause of IC/PBS remains unclear, with several potential factors having been proposed, including inflammation, mast cell activation, genetic predisposition, autoimmune mechanisms, and neurogenic factors [28,29]. These complexities make IC/PBS particularly difficult to study, highlighting the pressing need for new therapeutic strategies.

Some behavioral changes can complement pharmacological treatments in alleviating the symptoms of IC/PBS. Modifying urine volume and concentration through fluid restriction or increased hydration, avoiding certain bladder-irritant foods, and using supplements such as nutraceuticals, calcium glycerophosphates, and phenazopyridine are just some of the suggested strategies [3]. Additionally, appropriate manual physical therapy techniques—such as maneuvers that resolve pelvic, abdominal, and/or hip muscular trigger points; lengthen muscle contractures; and release painful scars and connective tissue restrictions—have proven to be particularly effective for patients presenting with pelvic floor tenderness [30].

Oral pharmacotherapy becomes the primary option when conservative approaches fail [30], whereas several intravesical instillation therapies can be applied as a second-line treatment, including chondroitin sulfate (CS), hyaluronic acid (HA), heparin, lidocaine, pentosan polysulfate sodium (PPS), and dimethyl sulfoxide (DMSO). Most of these therapies are designed to restore the structure of the glycosaminoglycan (GAG) layer, reinforcing its natural defense and barrier action [31]. In Europe, CS and HA are among the most prescribed agents. A typical regimen consists of one instillation per week for six weeks, followed by monthly instillations as needed. Gülpınar et al. randomly selected 42 patients for receiving either CS or HA. After six months, both agents significantly reduced pain (*p* < 0.001). However, CS demonstrated superior results in decreasing 24 h frequency (*p* < 0.001) and was the only agent to significantly improve nocturia (*p* < 0.001) [32].

A subsequent study investigated the outcomes of combined HA/CS instillation. Özkıdık et al. randomly selected 72 patients for receiving either HA, CS, or a combination of HA and CS, following them over a 24-month treatment period. The greatest reduction in pain was observed in the HA/CS group, although this outcome was not statistically significant compared to the other treatment arms (*p* = 0.15). However, improvements in both urgency and Health-Related Quality of Life (HRQoL) scores were significantly better in the HA/CS group (*p* = 0.04 and *p* = 0.02, respectively) [33].

Overall, these treatments are safe and well-tolerated; the most frequently reported adverse events include pain, irritation, and urinary tract infections (UTIs). To the best of our knowledge, there is no strong evidence suggesting that any particular instillation therapy is superior to any of the others [31]; however, PPS and DMSO present worse side effect profiles, including headaches and dizziness, and necessitate ophthalmologic monitoring due to the risk of lens opacification [10].

Recently, an innovative molecule was tested for bladder instillation: Adelmidrol, a synthetic derivative of azelaic acid that acts as a palmitoylethanolamide (PEA) enhancer, increasing endogenous PEA levels and exerting significant anti-inflammatory and immunomodulatory effects [34]. Like PEA, Adelmidrol belongs to the family of Autacoid Local Injury Antagonist Amides (ALIAmides). Its amphiphilic and amphipathic properties make it particularly soluble and suitable for topical and intra-articular administration [35,36]. Its efficacy has been demonstrated in numerous experimental studies on inflammatory systemic disorders. Di Paola et al. observed that a combination of hyaluronic acid and Adelmidrol reduced mechanical allodynia and motor dysfunction as well as articular cartilage degeneration in a murine model of osteoarthritis [37]. Furthermore, Cordaro et al. proposed using Adelmidrol as a new pharmacological approach to treating inflammatory bowel disease. In their study, Adelmidrol (10 mg/kg daily, administered orally) was tested in a murine model of colitis, and the outcomes were promising, as evidenced by a significant reduction in diarrhea, body weight loss, and levels of pro-inflammatory cytokines expressed in the colon [38].

In light of these encouraging results, in the present study, we aimed to investigate the anti-inflammatory effects of intravesical Vessilen^®^ (a formulation consisting of 2% Adelmidrol + 0.1% sodium hyaluronate) administration in patients affected by IC/PBS or other related conditions characterized by local inflammation, voiding dysfunctions, and pelvic/perineal pain. We observed a significant reduction in pain, urgency, and frequency intensity after a weekly treatment for 6 weeks, as demonstrated by the ICIQ-FLUTS score (89.3 vs. 61.3; *p* = 0.021) and VAS score (4.4 vs. 2.6; *p* < 0.001). The improvement was also confirmed by the personal satisfaction of most of the patients, with the mean PGI-I score being ≤3. No significant adverse reactions were noted during the treatment cycle, and only one patient interrupted the procedure due to urinary catheter discomfort.

Our findings align with the results of a previous observational study in which intravesical Vessilen was tested in a rodent model of IC/PBS induced by cyclophosphamide (CYP). In their paper, the authors concluded that intravesical treatment with Vessilen^®^ effectively reduced CYP-induced inflammation and pain by inhibiting the nuclear factor-κB pathway, lowering levels of inflammatory mediators, and alleviating mechanical allodynia while reducing nerve growth factor levels [39]. In a second step, these preclinical murine results were confirmed in a population of 128 men and women suffering from chronic pelvic pain associated with symptoms of the lower urinary tract. In line with our findings, symptom improvement (regarding pain, frequency, voiding dysfunction, and urgency) was reported after weekly treatment for 8 weeks [39].

In a more recent study, a combination of Adelmidrol and HA was administered to patients with non-muscle-invasive bladder cancer (NMIBC) after they underwent transurethral surgical resection (TUR). The use of Adelmidrol and HA intravesical instillation as a treatment supplementary to intravesical anticancer therapy helped control pain intensity, urgency, and frequent micturition-related discomfort, enabling all the patients to complete the entire course of anticancer treatment, including immunotherapy with Bacillus Calmette–Guérin (BCG) and intravesical chemotherapy with mitomycin C (MMC) or epirubicin (EPI). Moreover, continued treatment with Adelmidrol + HA after the anticancer cycle further improved symptomatology and decreased the side effects of chemotherapy [40].

This study represents the first attempt to test Vessilen in a homogeneous group of women affected by bladder-related chronic inflammatory conditions. There are certainly some limitations, namely, the small sample size and the lack of histological evaluation, with the results based solely on subjective questionnaires. However, future studies with greater patient enrollment and follow-ups will allow us to better determine the exact number of sessions required to achieve a clinical benefit, how long this benefit lasts, and whether maintenance cycles will be necessary.

## 5. Conclusions

This study demonstrates that intravesical administration of adelmidrol combined with sodium hyaluronate (Vessilen^®^) may offer significant anti-inflammatory benefits for patients with interstitial cystitis/bladder pain syndrome (IC/BPS) or other chronic bladder disorders, particularly in terms of alleviating pain and improving lower-urinary-tract symptoms. Despite the small sample size, this preliminary study shows that Vessilen^®^ is a safe, promising, and effective alternative for patients who cannot obtain sufficient relief from traditional oral pharmacological treatments. However, further research with larger patient cohorts is needed to determine the optimal treatment regimen and the number of sessions required to sustain these benefits over time.

## Figures and Tables

**Figure 1 healthcare-13-01340-f001:**
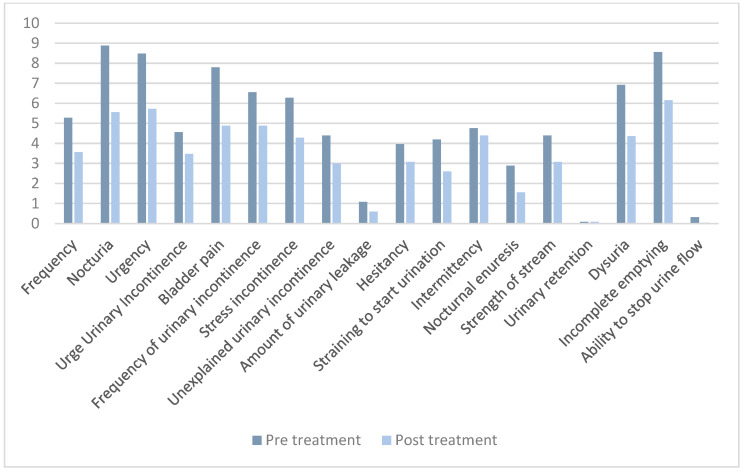
ICIQ-FLUTS scores distributed across 18 items.

**Table 1 healthcare-13-01340-t001:** Population’s baseline characteristics.

Population Characteristics	Value
Age (Years)	58.4 (16.4)
Multiparous (%)	22 (84.6%)
Smoking (%)	1 (3.8%)
Previous pelvic surgery (%)	11 (42.3%)
Increased urinary frequency (%)	14 (56%)
Urgency (%)	8 (32%)
Pelvic pressure/discomfort (%)	4 (16%)

**Table 2 healthcare-13-01340-t002:** Mean scores (standard deviation) of the pre- and post-treatment questionnaires. N/A: not applicable. Values in bold indicate statistical significance (*p* < 0.05).

Questionnaire	Pre-Treatment Score	Post-Treatment Score	*p*-Value
Total ICIQ-FLUTS	89.3 (41.1)	61.3 (42.5)	**0.021**
- Frequency	5.3 (5.1)	3.6 (4.3)	**0.009**
- Nocturia	8.9 (4.5)	5.6(4.8)	**<0.001**
- Urgency	8.5 (4.0)	5.7 (3.9)	**<0.001**
- Urge urinary - incontinence	4.6 (5.1)	3.5 (4.2)	0.061
- Bladder pain	7.8 (4.5)	4.9 (4.0)	**<0.001**
- Frequency of urinary incontinence	6.6 (5.7)	4.9 (5.2)	**0.022**
- Stress incontinence	6.3 (4.8)	4.3 (4.3)	**0.006**
- Unexplained urinary incontinence	4.4 (5.0)	3.0 (4.4)	**0.007**
- Amount of urinary leakage	1.1 (1.1)	0.6 (0.6)	**0.006**
- Hesitancy	4.0 (4.2)	3.1(4.0)	**0.013**
- Straining to start urination	4.2 (4.8)	2.6 (3.8)	**0.007**
- Intermittency	4.8 (4.5)	4.4 (4.5)	0.269
- Nocturnal enuresis	2.8 (4.4)	1.6 (3.1)	**0.027**
- Strength of stream	4.4 (4.1)	3.1 (3.9)	**0.021**
- Urinary retention	0.1 (0.4)	0.1 (0.4)	0.500
- Dysuria	6.9 (4.8)	4.4 (4.1)	**0.002**
- Incomplete emptying	8.6 (4.6)	6.2 (4.2)	**<0.001**
- Ability to stop urine flow	0.3 (0.6)	0.04 (0.2)	**0.008**
VAS	4.4 (1.5)	2.6 (1.8)	**<0.001**
PGI-I	N/A	2.8	N/A

## Data Availability

The data presented in this study are available on request from the corresponding author.

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
