# Peer review of "Exploring the Efficacy of Vessilen® in Treating Bladder Pain Syndrome/Interstitial Cystitis: A Prospective Study"

_healthcare, 2025, doi:10.3390/healthcare13111340_

Round 1
Reviewer 1 Report
Comments and Suggestions for Authors
1-I recommend to use the PBS (Painful bladder syndrome) instead of BPS.
2- You mentioned that "This was a prospective observational study".It seems you should designed this study as a RCT and compare the results with standard available treatments.
3- The introduction part include redundant data and should be changed. The unnecessary data should be revised.
4-Please refine the exact type of study.
5-The demographic data of patients are not complete, please add dominant symptoms, presence of cystoscopic finding ,etc....
Author Response
Response 1: I replaced all the acronyms BPS in text with PBS (Painful Bladder Syndrome).
Response 2-4: I defined the study as a pilot study due to the small sample size, the rarity of the condition and the impossibility of having a proper control arm.
Response 3: I shortened the introduction by removing redundant information
Response 5: In the population characteristics, I included the main reported symptoms; however, we did not include cystoscopic findings, as cystoscopy is no longer required according to the most recent diagnostic definitions.
Reviewer 2 Report
Comments and Suggestions for Authors
The subject is novel and interesting. However it concerns disease which is not common, so results are hard to compare and it is hard to broaden the group of included patients.
Manuscript is written in a professional way, with a deep and profound scientific backgroung.
- I suggest "preliminary study", not prospective study - due to low number of patients and lack of longer follow - up results. I am sure that results after few months/ year would enhance importance of the study (which might be suggested in a conclusion).
- Number and date of Ethic Committee should be included.
- Statistics are made the proper way, but along with the results they are not satisfactory. I suggest to split the results of ICIQ-FLUTS into separate items and show the importance of changes (tables or charts). It would also provide informations about patients complaints before the procedure.
- Exact method of procedure should be described. Also, informations about test made before the operations should be included. In future, ultrasound with post-void residual urine volume should be included.
- Introduction is very long compared to the discussion. Provide just the most important informations at the beginning.
- Current diagnosis criteria should be presented in the table or in the text to make readers understand it better. Also, if 9 months time is not a criterium anymore, please note that - because your inclusion time criterium was shorter.
Author Response
Response 1: I defined the study as a pilot study due to the small sample size, the rarity of the condition, and the impossibility of having a traditional control arm.
Response 2: All the required details regarding ethics committee approval have been included in the text.
Response 3: I divided the ICIQ-FLUTS into its 18 items, comparing the scores before and after treatment for each of them. I presented these data in a table and a histogram.
Response 4:I provided a detailed explanation of how to perform the treatment and the required evaluation before the therapy.
Response 5:I shortened the introduction by removing redundant information.
Response 6: The definition of PBS has changed significantly over time, and there is still no unanimous consensus among the major urological societies. In particular, the most recent definitions are less restrictive regarding symptom duration and no longer require cystoscopy as a diagnostic criterion.
Round 2
Reviewer 1 Report
Comments and Suggestions for Authors
All issues are addressed appropriately